# Adolescents’ Long-Term Experiences of Manageability, Comprehensibility, and Meaningfulness of a Group-Based Exercise Intervention for Depression

**DOI:** 10.3390/ijerph19052894

**Published:** 2022-03-02

**Authors:** Sara Reinodt, Emma Haglund, Ann Bremander, Håkan Jarbin, Ingrid Larsson

**Affiliations:** 1School of Health and Welfare, Halmstad University, SE-30118 Halmstad, Sweden; sara@reinodt.com; 2Spenshult Research and Development Centre, SE-30274 Halmstad, Sweden; emma.haglund@hh.se (E.H.); ann.bremander@fou-spenshult.se (A.B.); 3Section of Rheumatology, Department of Clinical Sciences, Lund University, SE-22242 Lund, Sweden; 4Department of Environmental and Biosciences, School of Business, Innovation and Sustainability, Halmstad University, SE-30118 Halmstad, Sweden; 5Department of Regional Health Research, University of Southern Denmark, DK-5230 Odense, Denmark; 6Danish Hospital for Rheumatic Diseases, University Hospital of Southern Denmark, DK-6400 Sonderborg, Denmark; 7Child and Adolescent Psychiatry, Department of Clinical Sciences Lund, Lund University, SE-22184 Lund, Sweden; hakan.jarbin@regionhalland.se; 8Child and Adolescent Psychiatry, Region Halland, SE-30185 Halmstad, Sweden; 9Department of Health and Care, School of Health and Welfare, Halmstad University, SE-30118 Halmstad, Sweden

**Keywords:** adolescents, comprehensibility, depression, exercise intervention, manageability, meaningfulness, qualitative content analysis, treatment

## Abstract

Physical exercise is a potentially effective treatment for adolescents with mild to moderate depression. However, there is a lack of long-term follow-ups to reveal adolescents’ experiences of exercise as a treatment for depression. The salutogenic concept of sense of coherence (SOC), comprising the domains manageability, comprehensibility, and meaningfulness is important to understand behaviour change. This study aimed to describe adolescents’ long-term experiences of manageability, comprehensibility, and meaningfulness of a group-based exercise intervention for depression. Fourteen adolescents with persistent depression were recruited from a psychiatric outpatient clinic and interviewed one year after participating in a 14-week moderate to vigorous exercise intervention for depression. An abductive qualitative content analysis was conducted, based on the three SOC domains manageability, comprehensibility, and meaningfulness. The results revealed that participation in the intervention was made manageable by a supportive environment, including: the intervention design, togetherness with peer group, and encouragement from adults. The comprehensibility of the intervention emerged through the insights regarding health benefits of exercise and the aim of the intervention. Meaningfulness was achieved through improved health behaviour, well-being and self-esteem, along with strengthened belief in the future and increased commitment to everyday life. The group-based exercise intervention was experienced as manageable, comprehensible, and meaningful.

## 1. Introduction

Mental health disorders are common in adolescents, and depression is one of the most frequent diagnoses [1]. Depression is associated with functional impairment in daily life and an increased risk of suicide, and may severely affect educational and social achievements [2,3]. Depression is heterogeneous, which in turn calls for a range of interventions [2,4,5]. Currently, available treatment interventions for depression in adolescents include pharmacological treatment and psychotherapy, which have shown modest efficacy [6,7]. Exercise as a stand-alone treatment or as an add-on treatment for depression in adolescents has not yet been adequately investigated, but research shows promising results, with positive effects on depressive symptoms and overall well-being [8,9,10,11,12]. In adults with depression, aerobic exercise of moderate to vigorous intensity has shown effects equal to medication or psychotherapy and is recommended by the European Psychiatric Association [11]. However, because studies of exercise in adolescents [9,10,13,14,15] have less than adequate quality of methodology, use wait-list control groups, and involve disparate recruitment process and exercise doses, properly controlled studies with clinically sound recruitment and long-term follow-ups are required [11,15].

To better understand the outcome of exercise intervention studies and to further develop effective exercise interventions for adolescents with depression, it is important to gain a better understanding of how adolescents experience exercise regimes [8,16]. People with depression who are invited to participate in exercise interventions often change their health behaviour, which can be challenging [8,17]. After completing an exercise intervention aimed at lowering depressive symptoms, many adolescents describe positive changes in their attitudes to exercise and in relation to their exercise habits [8,18]. Qualitative studies describing long-term evaluations of health behaviour after an exercise intervention for adolescent depression are scarce, but Sunesson et al. (2021) identified improved self-esteem and a supportive environment as facilitators in the maintenance of exercise one year after participating in the intervention, while disease burden and a lack of a supportive environment were identified as barriers [19].

Health-promoting interventions that encourage behaviour change, such as maintenance of exercise habits, need to focus on people’s abilities and resources and align with the salutogenic approach [20,21]. Sense of coherence (SOC) is the main concept in the salutogenic approach to health. SOC includes three components: (1) manageability, which refers to the capacity to cope and use available resources to cope with challenges, (2) comprehensibility, the cognitive dimension, which refers to external and internal stimuli that are perceived as structured, predictable, and explicable, and (3) meaningfulness, the motivational dimension, which is about the challenges a person meets being worthy of investment and engagement [22,23]. The three dimensions of SOC are interrelated and seem to be strongly integrated, implying that the management of stressors is dependent on all three [22]. SOC is related to health in general and mental health in particular. SOC contributes to the understanding of elements for health [24] and is associated with improved health behaviour [20,24,25].

In order to understand how to encourage adolescents to maintain exercise habits after an exercise intervention among adolescents with depression, it is important to understand how adolescents experience the manageability, comprehensibility, and meaningfulness of physical exercise. It can therefore be argued that a study of long-term experiences of an exercise intervention within the framework of SOC may provide valuable information for the future development of effective exercise treatment for adolescents with depression. Thus, the aim of the present study was to describe adolescents’ long-term experiences of manageability, comprehensibility, and meaningfulness of a group-based exercise intervention for depression.

## 2. Materials and Methods

### 2.1. Study Design

The present study had a descriptive, qualitative design. A qualitative content analysis was chosen to reveal the variation and diversity in the text material [26]. An abductive approach was used to describe the adolescents’ long-term experiences of a group-based exercise intervention for depression, which allowed the analysis to move back and forth between inductive and deductive qualitative content analysis, to arrive at a more complete understanding [27]. The three SOC domains were used as a framework for the deductive part of the analysis [22]. The study conforms to the consolidated criteria for reporting qualitative research (COREQ) [28].

### 2.2. Setting

The interviews were conducted at 12-month follow-up of an intervention study aimed to test feasibility, acceptability, and preliminary effects of group-based aerobic exercise as a treatment for depression in clinically referred adolescents [29]. Adolescents were recruited at an outpatient child and adolescent psychiatry (CAP) clinic in the south of Sweden. They were diagnosed with mild to moderate persistent depression (duration 1.4–5.3 years, median 2.2 years) according to the DSM-5 criteria. The participants also suffered from significant comorbidities, such as ADHD, anxiety syndromes, and PTSD, as described in detail elsewhere [18,29]. All participants reported an exercise level below 150 min/week moderate exercise or 75 min/week intensive exercise prior to the intervention. Further inclusion criteria, as defined by the intervention study, [29], were: currently under treatment at a CAP clinic, and previously participated in basic psychosocial treatment without improvement. Exclusion criteria were having a current eating disorder, intellectual disability or autism precluding group participation, need of an interpreter, high risk of suicide, chaotic social circumstances precluding a regular exercise schedule, ongoing psychotherapy, or medication adjustment within the past four weeks, or having a chronic illness proscribing vigorous exercise.

The adolescents participated in a 14-week exercise intervention, described elsewhere [29]. The group sessions were of 60 min, three times weekly, and included both aerobic and strength exercises. The aerobic level was at least moderate (more than 70% of maximum heart rate) for 32 min (range 18–40) and vigorous (more than 80% of maximum heart rate) for 17 min (range 6.5–27). Adherence to the 36 sessions was 75% (range 19–97%). During the intervention, each participant had three individual coaching sessions with a personal trainer, to receive personalised recommendations and evaluation of the exercise. Further, the study coordinator supported the participants and their families with reminders, pep talks, and personal involvement in the group sessions, with the purpose of facilitating adherence.

### 2.3. Participants

Sixteen adolescents completed the exercise intervention and were invited to participate in the one-year follow-up interviews. Fourteen adolescents accepted the invitation and were interviewed. The participants were ten females and four males, aged 14–19, median disease duration at baseline at 2.2 years and median Body Mass Index of 28.6. At the one-year follow-up after completing the exercise intervention, all boys (*n* = 4) and four girls reached disease remission (Table 1).

### 2.4. Data Collection

Individual, semi-structured follow-up interviews were conducted during March and April 2019 and took place in a private room at the CAP clinic; the staff were competent to address any feelings that arose during the interview. The interviews were performed by one of the researchers, a nurse (IL), who had no previous relationship with the participants. The interviewer has extensive experience of conducting qualitative interviews and endeavoured to be continually conscious of keeping an open mind and listening to the participants, by asking open-ended questions and by prompting the participants to develop their narrative. It was optional for the adolescent to have a parent present in a passive role during the interview.

The interview guide included an initial question: “Can you describe how you experienced the training you attended a year ago?” Questions followed regarding their experience of exercise since the intervention ended: “Can you describe what exercise means for you?” and “What opportunities have you had to continue exercising after the intervention?” The interviews also included questions regarding their perceptions about exercise as treatment and effects experienced from the exercise, for example: “Can you describe in what way you experience that exercise affects your well-being?” and “What are your perceptions about exercise as a treatment for depression?” The adolescents were encouraged to develop their answers by asking them follow-up questions, such as: “Please, tell me more about…” and “What do you have in mind when you say…?” To assess the interview guide, two pilot interviews were conducted. As no amendment was required, these interviews were included in the study. The digitally recorded interviews lasted from 21 to 46 min, with a median of 36 min and a total interview length of eight hours and nine minutes.

### 2.5. Data Analysis

The abductive qualitative content analysis started with an inductive part, in which the transcribed interviews were listened to and read through several times, to get a sense of the whole [27]. Sentences relevant to the aim were identified as meaning units by the first author (SR). A total of 212 meaning units emerged from the transcribed interviews. Condensation of the meaning units, with the aim of shortening the text while still retaining the content, was then performed. The condensed meaning units were abstracted into codes to describe how the adolescents experienced the exercise intervention. During this process, the whole context was considered, and the process of the analysis included a movement between the whole and parts of the text [26]. The codes were then compared and codes with similar contexts were sorted into sub-categories, with the aim that no data should fall between the sub-categories or fit into more than one sub-category. Based on their similarities and differences, the sub-categories were then sorted into categories. Nine sub-categories and four categories emerged, which formed the manifest content. The deductive part of the analysis meant that, based on their content, the categories were sorted into the three SOC domains: manageability, comprehensibility, or meaningfulness [22,27]. The analysis was continuously discussed between the authors SR, EH, and IL and revised several times. When consensus was reached between all authors, all four categories had been linked to one of the three SOC domains. To increase trustworthiness, the authors, who had extensive experience in paediatric psychiatry (MD, PhD), nursing (RN, PhD), physiotherapy (PT, PhD) and qualitative methodology, respectively, participated in both the research design and data analysis process.

### 2.6. Ethical Considerations

The Regional Ethical Review Board at Lund University, Sweden approved the study (No. 2017/98). The study was conducted in accordance with the ethical principles of the Helsinki Declaration [31]. The adolescents were provided both oral and written information about the study and could withdraw at any time without giving a reason and with no consequences for their treatment at CAP. The adolescents and their parents/legal guardians provided written informed consent prior to the intervention and before the one-year follow-up interviews. After the interview, participants were also given the opportunity to discuss with staff at the CAP clinic thoughts and emotions that had arisen during the interview.

## 3. Results

The adolescents’ long-term experiences of manageability, comprehensibility, and meaningfulness of a group-based exercise intervention consisted of four categories and nine subcategories (Table 2).

### 3.1. Manageability

The domain manageability included the category “a supportive environment made the exercise intervention manageable”, which comprised the subcategories: the intervention design, experiencing togetherness with peers in a group, and experiencing encouragement from adults.

#### 3.1.1. A Supportive Environment Made the Exercise Intervention Manageable

A supportive environment, provided by the study design, made the exercise intervention manageable for the adolescents. Experiencing togetherness with peers in a group, together with support and encouragement from adults, contributed to a sense of security in managing the group-based exercise.

##### The Intervention Design

The design of the intervention brought structure to exercising and made the participation manageable. Recurring weekly exercise sessions provided a consistent routine for the adolescents and supported the manageability of participating in a group-based exercise intervention.
“It was just like a routine. That is because I wrote down all the dates and then my parents gave me a lift there. So, it became a routine. It was like a practice, these two or three days I went there. So, it worked well.”(Adolescent No. 9)

The participants found that the exercise was both physically and mentally feasible, which can be interpreted as the design of the intervention was manageable. On the other hand, some of the adolescents experienced that the initial exercise intensity was too low and appreciated when the intensity became more challenging. The structure of the group sessions was perceived as clear and understanding the structure of the sessions was described as comforting. Thus, the structure of the exercise sessions made the intervention manageable, although all activities were not always fun. Some participants expressed that being able to choose more individually tailored exercises would have been appreciated.
“Well, it wasn’t fun that somebody told me what to do all the time. That I was not allowed do what I wanted to do. And then you were supposed to do something else, which was not as fun.”(Adolescent No. 14)

##### Experiencing Togetherness with Peers in a Group

Manageability was facilitated by feelings of togetherness experienced by the adolescents during the exercise intervention. The participants described the challenging experience of being introduced to a group of new people. However, during the intervention, they got to know each other, which made group sessions a safe space. The size of the group contributed to the feeling of a supportive environment surrounding the adolescents. Exercise in a group with peers who also suffered from depression led to feelings of togetherness, thereby creating a connection within the group.
“I have always felt that it is challenging to exercise in a group. And now we did this in a group where I knew that all of us were the same and had anxiety just like me and also suffered from depression and then it was a little bit easier because it was people who actually understood.”(Adolescent No. 10)

Feelings of togetherness in the group led to the sessions being perceived as a place to have fun with peers. The adolescents described a supportive environment where they dared to challenge themselves during exercises, without feelings of being judged.

##### Experiencing Encouragement from Adults

The participants felt supported and encouraged by the adults around them throughout the intervention, which made the intervention manageable. They described how the supportive environment in the exercise setting made them feel encouraged when they showed up and completed each session, even on days with low energy or bad mood.
“The personal trainer kind of coached you in a good way, you felt safe there. You didn’t feel any pressure, but you didn’t feel that you should sit and take it easy either.”(Adolescent No. 1) 

The personal trainer encouraged the adolescents to participate in the sessions for their own sake, and they experienced that it was not all about performing. Feelings of having control over their own exercise increased the sense of manageability. The participants felt encouraged but never pressed to try new things, which supported them to dare to push their limits. For some participants, supportive dialogues with their families helped them to notice gradual improvements in mood and well-being and changes in daily life routines.
“Because it is so small [changes], you barely notice on your own, because it happens sort of progressively, which makes you think that this is how I usually am.”(Adolescent No. 2)

The participants also experienced support and encouragement from the school to adapt their weekly schedule, which made participation in the exercise intervention manageable.

### 3.2. Comprehensibility

The domain comprehensibility included the category “the emerging insights made the exercise intervention comprehensible”, comprising the subcategories understanding health benefits of exercise and understanding the aim of the intervention.

#### 3.2.1. The Emerging Insights Made the Exercise Intervention Comprehensible

Insights that emerged during the group-based exercise intervention made participation comprehensible for the adolescents. The insights included an understanding of the health benefits of group-based aerobic exercise as a treatment for depression. Another insight that contributed to the comprehensibility of the exercise intervention was when the adolescents understood that the aim of the intervention was as a treatment option for their depression.

##### Understanding Health Benefits of Exercise

The participants experienced that their participation in the intervention provided them with new knowledge about exercise, e.g., how to reduce the risk of pain or injuries. A better understanding of various ways to perform exercise and insights about beneficial outcomes on health and well-being made the intervention comprehensible. These insights contributed to the understanding of the importance of participating in physical education classes in general life. But, even if they got a better understanding of how to exercise for beneficial effects, it could still be challenging to adhere to the recommended exercise frequency and intensity when they were on their own. Thus, exercising on their own could lead to less effect on well-being, in comparison to the effects achieved during the exercise intervention.
“But then I thought, I want to go to the training because it is right now I need it the most. So, this week I have attended and even though it has been tough to get up and go there, I have felt that I need it and that it helps me.”(Adolescent No. 4)

##### Understanding the Aim of the Intervention

The participants described the importance of a clearly stated aim of the intervention to make it comprehensible. The intervention was perceived as interesting and the purpose of collecting research data made it a comprehensible experience. The opportunity to try a new treatment method without side effects was appreciated. Such insights contributed to comprehensible participation, including for those adolescents who had not experienced any significant effects on their mental well-being.
“I understand the idea behind it and that it might work, but it didn’t really work for me. But I was not really surprised either, because I have tried almost everything, and it has not worked. So, psychologically I cannot remember that I got as much out of it as physically…”(Adolescent No. 1)

### 3.3. Meaningfulness

The domain meaningfulness included two categories and four subcategories. The first category “improved health behaviour made the exercise intervention meaningful”, comprised the subcategories: experiencing increased well-being and improvement of daily routines and experiencing improved self-esteem. The second category, “a strengthened belief in the future made the exercise intervention meaningful”, comprised the subcategories: experiencing increased hope for the future and experiencing increased commitment to everyday life.

#### 3.3.1. Improved Health Behaviour Made the Exercise Intervention Meaningful

Experiencing improvements in health in terms of increased well-being led to improved routines in everyday life, which made the exercise intervention meaningful. Further, experiencing improved self-view after participation in the group-based exercise contributed to improved health behaviour and made the intervention meaningful.

##### Experiencing Increased Well-Being and Improvement in Daily Routines

The participants experienced increased well-being, with added physical and mental energy. They described the exercise intervention as meaningful as it contributed to feelings of happiness, strength, and well-being. Participation in exercise was described as a fun and positive experience, which made attendance meaningful. Some participants experienced short-term positive effects on their health and well-being that were limited to the time immediately after an exercise session but faded out the next day. Regular exercise during the intervention period resulted in improved routines regarding food and sleep. For example, it was easier to fall asleep and they slept better the night after an exercise session.
“I had more energy, and I could even fall asleep in the evening. Well, I just felt that everything became so much clearer in all possible ways.”(Adolescent No. 12)

The participants experienced that their improved well-being facilitated school attendance and their attention span, making everyday life easier. They experienced a feeling of being more clear-headed, which increased their focus and ability to participate in schoolwork. All this, together with the absence of stressful thoughts during and after the exercise sessions made the intervention meaningful. The adolescents expressed that exercise served as a tool to reduce feelings of restlessness, anger, frustration, and mood swings. Exercising enabled time for reflection and processing of thoughts.
“Afterwards, I was only sad in a normal way and because of usual reasons. So, it made me feel well.”(Adolescent No. 7)

The participants pointed out that, even though they experienced the exercise as meaningful because of its effects on well-being, the exercise was not experienced as a universal solution. In addition, they indicated a need to process and manage emotions and painful events through counselling.

##### Experiencing Improved Self-Esteem

Experiences of improved self-esteem also made the intervention meaningful. Participation gave them the chance to practice challenging themselves while exercising, without fear of failure, which created a sense of achievement. The participants described how this enhanced their self-confidence and self-esteem, which in turn had a positive impact on their well-being. It could sometimes be hard to describe the progressive changes in health behaviour and well-being during the preceding year.
“Well, I know that I feel a big difference, but I don’t know what has changed. But I do not feel like the same person, I have more will power now, I know what can happen afterwards [the feeling after exercise], I know how… I don’t know. I know how I was before and I know how I am now, and it is such a big difference, but I am not sure exactly what has changed.”(Adolescent No. 6)

The participants often experienced pride in being able to manage the exercise intervention, which strengthened their self-esteem through the feeling of doing something good for their body and mind. Experiencing the progress of becoming stronger and having more stamina during exercise made participation meaningful. This also facilitated an improved view of one’s own body, through a changed self-image and a feeling of being good enough. Furthermore, their self-esteem was strengthened by a shift of focus on well-being instead of on performance.
“I think that, during this last year, the exercise has been very much for myself, for my own well-being. Not like before, exercising to satisfy others or to feel good about myself by competing with others. While this year has really been about what I enjoy, what I am good at, and what I like to do for me, and not for anybody else. That’s the big difference.”(Adolescent No. 5)

Transforming exercise into a meaningful way to challenge themselves without feeling stressed resulted in improvement of health behaviour.

#### 3.3.2. A Strengthened Belief in the Future Made the Exercise Intervention Meaningful

A strengthened belief in the future was one outcome of the exercise intervention, which made the intervention meaningful for the adolescents. Their strengthened belief in the future was based on a hope of regaining health and on experiencing a new treatment tool for a potential future depression or if the depression persisted. The strengthened belief in the future was also founded in increased commitment to everyday life and leisure activities, which also contributed to the meaningfulness of the intervention.

##### Experiencing Increased Hope for the Future

Hope for the future was raised as early as when the adolescents were given the opportunity to participate in the intervention. Although some participants expressed that, initially, they had been skeptical towards participating, they were willing to try a new treatment for their depression. Feelings of curiosity and hope about what effects the exercise could have for their well-being were also described.
“Well, I have been thinking about it for a long time, I do like to be physically active. It is just that I sort of need motivation and a reason to do it. So, when they asked me I was like ‘yes, I would actually like to join. I think it can do something; I want to see if it makes any difference.’ I want to start exercising, I really do. To feel strong and healthier. So, well, I was pretty thrilled to do it actually.”(Adolescent No. 12)

The participants enjoyed participating in the group-based exercise. During the intervention, they sometimes longed for the next session, contributing to a belief in the future. The adolescents welcomed this, as such feelings had not been experienced for a while. They looked forward to continuing to exercise on their own, after completing the intervention, with an expectation of additional effects on their health and well-being. Expectations of eventually being able to discontinue medication made participation meaningful. The exercise intervention was described as providing a meaningful tool, to gain control over their well-being, both at that time and in the future.
“After I did the exercise with the personal trainer, I took so much from it that feel that I can do it [exercise] myself now.”(Adolescent No. 6)

##### Experiencing Increased Commitment to Everyday Life

The exercise intervention was meaningful as it contributed to a commitment to leisure activities, such as sports, meeting friends, or continuing exercising. Some of the participants found new activities to engage in, while others found their way back to previously enjoyed activities. Experiencing the benefits of a more active lifestyle increased a belief in the future, which had begun to grow during the intervention. Finding an individual exercise routine or a better balance between being active or sedentary during leisure time facilitated commitment and meaningfulness in exercise for the adolescents. Those who did not prefer to continue exercising in the same way as during the intervention felt inspired to try other kinds of physical activities.
“I do actually believe that exercising for your own health when you don’t feel good is not very common, mainly because when you don’t feel good or are depressed you don’t want to do anything and then you don’t start to exercise. But if you are forced to do it, and get into the routine it will help, because then you begin to do things. So, it helps an awful lot, it does.”(Adolescent No. 7)

In addition to being introduced to an exercise routine, the intervention design was experienced as meaningful, and therefore created commitment among the adolescents. The overall experiences during the exercise intervention contributed to a stronger belief in the future, which the adolescents had not felt for a long time.

## 4. Discussion

We found that the adolescents experienced manageability, comprehensibility, and meaningfulness of the intervention, which facilitated adherence and changes in health behaviour. We found that the three domains are strongly connected, and all of them are important when aiming to improve adolescents’ physical and mental health [21,25]. It might therefore be beneficial, when planning an exercise intervention as a treatment for depression, to target factors that strengthen SOC. The adolescents’ experiences from the intervention related to SOC can be summarized as a manageable setting for the intervention, comprehensible insights, and meaningful outcomes regarding health behaviour. These factors together contribute to a sense of coherence, which is important to creating a positive experience of a group-based exercise intervention and facilitates exercise participation [21].

In the present study, adolescents performed a graded exercise intervention, in which the intensity progressed from moderate to vigorous, in accordance with prevailing evidence [11,32]. Heart rate monitors were used to ensure the progressive increase in intensity over the weeks. The graded exercise intervention was well-received by the adolescents, and they described a sense of achievement. The adolescents’ ability to cope with exercise at this intensity is an interesting finding, given that the adolescents were burdened with long-term depression and psychiatric comorbidities [29]. Previous research has shown that to improve outcomes of depression in adults, aerobic exercise of moderate to vigorous intensity, two to three times a week, shows effects that are equal to those achieved by medication or psychotherapy [11]. Therefore, moderate to vigorous exercise has been recommended because of its tendency to have further effects on depression in addition to the more known effects on cardiorespiratory fitness [32]. The importance of vigorous-intensity exercise is also supported by another clinical study, in which adolescents with depressive symptoms were encouraged to exercise at their preferred intensity, which transpired to be at a very low-intensity level. The results of that clinical study showed no effect in reducing depression in the short term, but some effect after six months [33]. Our findings suggest that a graded exercise intervention of moderate to vigorous intensity, in accordance with recommendations for exercise treatment in depression [15], was well tolerated in the current clinically recruited sample of adolescents with significant morbidity.

The *manageability* of the intervention was based on a supportive environment, where supportive and encouraging adults and the feeling of belonging to a peer group were important. Support from those around the teenagers, such as family, healthcare professionals, coaches, and fellow group members, was important and facilitated adherence and commitment to the exercise intervention [18,19]. In the present study, extensive support from health professionals, with reminders, phone calls, pep talks, and personal participation in sessions, together with family support, was of utmost importance for compliance, albeit it might be challenging to implement in usual care [29]. This is congruent with research that shows that parents and significant adults have an important role to play in making an exercise intervention successful and manageable for adolescents [34]. Sharing training experiences with peers results in an understanding of how to act towards and respond to each other in the group, and just knowing other adolescents with similar difficulties is comforting [8]. By providing a setting for the exercise intervention that leads to a manageable, comprehensible, and meaningful experience for the adolescents, SOC can potentially be influenced in a positive direction and facilitate behaviour change for previously inactive adolescents with depression [20,21]. Exercising adolescents show a higher level of SOC in the dimensions of manageability and comprehensibility [35]. This may also influence the maintenance of exercise with lasting changes in health behaviour, such as improved routines for exercise, sleep, or diet, as described by the adolescents in our study.

The *comprehensibility* of the exercise intervention was based on insights among the adolescents with an understanding of the health benefits and thus the aim of the exercise. The beneficial results on the participants’ health were an encouraging and valuable experience and contributed to the comprehensibility. This is supported by previous research, which has suggested that exercise is a promising, feasible, and safe treatment strategy, and, with lower drop-out rates than psychotherapies. Exercise is well-accepted among adolescents with depression [10,15]. However, individuals with higher depressive scores at baseline tend to have higher drop-out rates, possibly influencing the outcome of the present results [36]. In our study, continuous support was required to promote adherence and achieve comprehensibility for continued exercise.

The *meaningfulness* of the exercise intervention was based on the adolescents’ experiences of the positive impact of exercise on daily life routines, such as sleep, diet, school attendance, and schoolwork. The experiences of increased energy and well-being remained one year after the exercise intervention stopped [18]. This is suggested as an important finding, given that meaningfulness, the motivational component of SOC, is thought to be accountable for the maintenance of manageability and comprehensibility [22]. This also appears to be in line with previous research [8], in which it is described that being in a “cycle of change”, where beneficial changes appear to influence further positive changes, e.g., exercise, increases self-confidence and thus contributes to improved mood and sleep, which in turn gives adolescents increased energy. A positive cycle of change increases motivation and meaningfulness for exercise [8]. Our findings of exercising at a vigorous level indicate an added dimension of self-efficacy and thus improved self-view, as participants managed to challenge themselves physically. Further, managing this increased degree of exertion possibly contributes to further meaningfulness, due to increased well-being. Exercise at a vigorous level for adolescents with depression conferred improved self-esteem and a feeling of being good enough, which is coherent with findings among healthy exercising adolescents [35]. Among adolescents, self-esteem and depression are linked, and experiences of low self-esteem may result in avoidance of challenging experiences, such as exercising [37]. Managing vigorous exercise can potentially have a positive impact on adolescents’ mental health, considering that a negative self-evaluation and feelings of worthlessness are some of the symptoms of depression [38]. Although the causal direction of the association between adolescents’ self-esteem and symptoms of depression is not yet established, the findings in our study are supported by a recent study that recommends that interventions for prevention and treatment of depression aim to facilitate a positive self-view [39].

The insight that exercise could work as a treatment for depression gave the adolescents in our present study hope for the future. They found a self-management strategy and a new way to manage their illness. Being able to self-manage depression creates a sense of empowerment, enhances confidence, and contributes to a sense of control, along with giving hope for the future. Self-management as exercise intervention empowers adolescents with depression to not only take more responsibility for their own recovery but also to take credit for it [40]. Empowerment is especially important for adolescents with depression because they often have difficulty seeing the bright side of life; they can view themselves as worthless, and lack belief in the future [41]. Social participation and commitment to everyday life increased among the adolescents in our study and contributed to the meaningfulness of the exercise. This is supported by Carter et al. (2016b) who described increased motivation towards engaging in homework and attending school, as well as increased socialization and communication with parents [8]. This is also an important finding in our present study, given that adolescents with depression often avoid social contact and isolate themselves by withdrawing from family and friends [41,42].

Trustworthiness in qualitative studies is often defined according to the four criteria of credibility, dependability, confirmability, and transferability [26,43]. Credibility was strengthened in our study by the transparent descriptions of data collection and data analysis, and there was a continuous discussion between the researchers during the analysis. The fact that two pilot interviews were conducted and that no new subcategories emerged after the eighth interview also strengthens the credibility. Dependability was strengthened by the fact that the same researcher conducted all the interviews, using an interview guide, and that the adolescents were encouraged to speak openly. To avoid influencing the adolescents to claim that the intervention was manageable, comprehensible, or meaningful, no questions were asked based on these concepts, which strengthens the dependability of the study. To create a safe environment for the interviews, the adolescents had the opportunity to have a parent present in the room, although most of them chose to be interviewed alone. The presence of parents can be both a strength and a weakness and might affect dependability. Regarding dependability, including more than one researcher in the analysis allowed for alternative interpretations of the results to be addressed. The researchers involved in the analysis had competence in qualitative methodology, physiotherapy, and child and adolescent psychiatry. The confirmability was strengthened in our study by the inclusion of numbered quotations from different participants, representing both positive and negative experiences. A further strength was that the qualitative analysis after one year reached 14 (88%) of the 16 participants who completed the intervention. Thus, participants with possibly less favourable experiences were not precluded from expressing views on the exercise intervention. However, two participants did not turn up to the first session of exercise and another one completed just five sessions. Their views were not covered but neither would they be relevant to a description of experiences of the exercise. A limitation could be the duration of the interviews; 21–46 min might be too short. However, the interview texts were considered rich in content and contained great variety [43]. The detailed description of the selection of participants together with the research process strengthened the transferability [27]. Regarding transferability, the presence of diagnoses other than depression, such as ADHD and anxiety disorders, is important to acknowledge, although such comorbidities are common among adolescents with depression [44]. Collecting all data at one clinic is another limitation. Providing that the adolescents in the present study are representative of adolescents with depression, their experiences of an exercise intervention could be transferable to a wider population.

## 5. Conclusions and Implications

This study contributes to adolescents’ long-term experiences of group-based moderate to vigorous exercise intervention for depression as manageable, comprehensible, and meaningful. The exercise intervention was experienced as manageable when the adolescents were surrounded by supportive people: exercising in a group with peers and receiving encouragement from adults. Emerging insights from the exercise intervention contributed to comprehensibility when the adolescents experienced health benefits and understood the aim of the intervention. The meaningfulness was represented by improved health behaviour: increased well-being and supporting routines in everyday life, along with improved self-esteem. Meaningfulness also comprised a strengthened belief in the future and hope of regaining health, which resulted in an increased commitment to and enjoyment in everyday life. The study suggests that group-based vigorous exercise as a treatment for depression can be valuable for adolescents in the long term. Further qualitative research is required to explore a range of ways of understanding the impact of exercise interventions as treatment for depression.

## Figures and Tables

**Table 1 ijerph-19-02894-t001:** Participant characteristics at one-year follow-up, after participating in group-based exercise intervention for depression (*n* = 14).

Characteristics	Participants (*n* = 14)	Female (*n* = 10)	Male (*n* = 4)
Age (years)			
Median (Min-max)	16.7 (14.0–19.0)	17.1 (15.7–19.0)	15.5 (14.0–17.5)
Disease duration at baseline (years)			
Median (Min-max)	2.2 (1.4–5.3)	2.0 (1.4–5.3)	2.9 (2.1–3.4)
Disease remission at one-year follow-up (*n*)	8	4	4
Body Mass Index (kg/m^2^)			
Median (Min-max)	28.6 (18.7–37.3)	29.6 (18.7–35.4)	22.4 (19.5–37.3)
Depression Score Clinician (QIDS-A_17_-C) *			
Median (Min-max)	4 (1–9)	4 (1–9)	4 (3–4)
Depression Score Self-rated (QIDS-A_17_-SR) *			
Median (Min-max)	6 (1–18)	10.5 (1–18)	5 (3–6)

***** QIDS-A_17_ (Quick Inventory of Depressive Symptomatology–Adolescent Version) Score: 6–10 points = Mild depression, 11–15 = Moderate, 16–20 = Severe, 21–27 = Very severe [30].

**Table 2 ijerph-19-02894-t002:** Overview of domains, categories, and subcategories that emerged in the qualitative content analysis of adolescents’ long-term experiences of manageability, comprehensibility, and meaningfulness of a group-based exercise intervention for depression.

Domain	Category	Subcategory
Manageability	A supportive environment made the exercise intervention manageable	The intervention design
		Experiencing togetherness with peers in a group
		Experiencing encouragement from adults
Comprehensibility	The emerging insights made the exercise intervention comprehensible	Understanding health benefits of exercise
		Understanding the aim of the intervention
Meaningfulness	An improved health behaviour made the exercise intervention meaningful	Experiencing increased well-being and improvement in daily routines
		Experiencing improved self-esteem
	A strengthened belief in the future made the exercise intervention meaningful	Experiencing increased hope for the future
		Experiencing increased commitment to everyday life

## Data Availability

Not applicable. The data will not be shared as ethics approval for the study requires that the transcribed interviews are kept in locked files, accessible only to the researchers.

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
