# Peer review of "Adolescents’ Long-Term Experiences of Manageability, Comprehensibility, and Meaningfulness of a Group-Based Exercise Intervention for Depression"

_ijerph, 2022, doi:10.3390/ijerph19052894_

Round 1
Reviewer 1 Report
It was a great pleasure that I reviewed the manuscript entitled “Adolescents’ long-term experiences of manageability, comprehensibility, and meaningfulness of a group-based exercise intervention for depression.” I think the paper some interesting findings. Here are several comments and suggestions that might help strengthen the paper. I am presenting those comments and suggestions in chronological order.
- In the Introduction section, the authors stated that because of “methodological flaws” in line 57. Please provide more details of flaws for cited references.
- In the second paragraph of the section of Setting (2.2), how come the authors did not provide the detailed of the intervention?
- The authors should provide the list of questions they used in Appendix or in Table for the section of Data Collection (2.3). Also, they need to discuss how they generated those questions and how those questions are trying to pertain 3 components.
- The Results section included 1-2 examples (meaning units) for each subcategory. However, the authors should state how many those units are classified into each subcategory. Also, the authors should describe how they were classified into each category (probably in the section of Data Collection).
- In the first paragraph of the Discussion section, the authors stated “a group-based exercise interventions for depression conformed well to three domains of SOC.” Unless there was priori criteria to classify meaning units into three domains before the data collection, I think this sentence is overstated because the study sounds like exploratory investigations on interviews. In addition, they stated that “… facilitated adherence and changes in health behaviour” and “…. strongly connected,” but how are they defined?
- Lines 466 to 468 are overstated. I don’t think the study found “improved health behavior” or “positive impact on daily life, routines such as sleep…”
- More generally, the authors should discuss how the study ensured a various factors (e.g., physical and mental health, major life change, etc.) after the intervention before the interview could influence the recruitment of participants and their responses for the interview.
Author Response
Reviewer 1
We thank the reviewer for a thorough review and helpful comments to improve the manuscript.
Please find the reviewers’ original comments and our response below (the latter using italics). In the revised manuscript, changes have been marked by track change to identify corrections.
It was a great pleasure that I reviewed the manuscript entitled “Adolescents’ long-term experiences of manageability, comprehensibility, and meaningfulness of a group-based exercise intervention for depression.” I think the paper some interesting findings. Here are several comments and suggestions that might help strengthen the paper. I am presenting those comments and suggestions in chronological order.
In the Introduction section, the authors stated that because of “methodological flaws” in line 57. Please provide more details of flaws for cited references.
Answer: We have added more details. Lines 58-60
In the second paragraph of the section of Setting (2.2), how come the authors did not provide the detailed of the intervention?
Answer: The components and the aerobic level of the intervention is described with more details. Lines 124-128
The authors should provide the list of questions they used in Appendix or in Table for the section of Data Collection (2.3). Also, they need to discuss how they generated those questions and how those questions are trying to pertain 3 components.
Answer: The questions are described in data collection. To avoid influencing the adolescents to claim that the intervention was manageable, comprehensible or meaningful, no questions were asked based on these concepts. Lines 154-165 + 522-536
The Results section included 1-2 examples (meaning units) for each subcategory. However, the authors should state how many those units are classified into each subcategory. Also, the authors should describe how they were classified into each category (probably in the section of Data Collection).
Answer: The total numbers of meaning units were 212 and is described in the Data analysis. The number of sentence meaning units in each subcategory is not considered relevant in qualitative studies as each meaningful carries equal weight whether it is one or more
In the first paragraph of the Discussion section, the authors stated “a group-based exercise interventions for depression conformed well to three domains of SOC.” Unless there was priori criteria to classify meaning units into three domains before the data collection, I think this sentence is overstated because the study sounds like exploratory investigations on interviews. In addition, they stated that “… facilitated adherence and changes in health behaviour” and “…. strongly connected,” but how are they defined?
Answer: We agree and have removed the first sentence and have adjusted the text in the paragraph Lines 420-426
Lines 466 to 468 are overstated. I don’t think the study found “improved health behavior” or “positive impact on daily life, routines such as sleep…”
Answer: We agree and have removed this
More generally, the authors should discuss how the study ensured a various factors (e.g., physical and mental health, major life change, etc.) after the intervention before the interview could influence the recruitment of participants and their responses for the interview.
Answer: We have added among strengths and limitations a comment about the degree of adherence to the interviews. Lines 545-550
Reviewer 2 Report
In view of the article reviewed and the references provided, this is a novel study in a field in which there does not seem to be much scientific literature, at least in medium- or long-term interventions.
This research provides some guidelines for planning physical exercise with persistent depression and significant comorbidities adolescents, in order to make its application manageable, understandable and meaningful.
The results section is well structured and very detailed. It offers examples of the responses of the adolescents participating in the study, which facilitates the qualitative understanding of the contributions of this research.
The authors incorporate 3 self-citations, which in view of the sample analysed and the time frame of the interventions, appear to derive from the same research project. However, in each of the 3 publications they carry out complementary and different analyses. They are relevant because they lend strength to the results reported in the manuscript.
The conclusions are presented in a concrete, clear and precise manner. The authors suggest that vigorous group exercise as a treatment for depression may be valuable for adolescents in the long term.
A specific section is included with strengths and limitations. For future research it would be interesting to have a larger sample to provide the results with greater scientific soundness.
Author Response
We thank the reviewer for a thorough review and helpful comments to improve the manuscript.
Please find the reviewers’ original comments and our response below (the latter using italics). In the revised manuscript, changes have been marked by track change to identify corrections.
In view of the article reviewed and the references provided, this is a novel study in a field in which there does not seem to be much scientific literature, at least in medium- or long-term interventions.
Answer: Thank you
This research provides some guidelines for planning physical exercise with persistent depression and significant comorbidities adolescents, in order to make its application manageable, understandable and meaningful.
Answer: Thank you
The results section is well structured and very detailed. It offers examples of the responses of the adolescents participating in the study, which facilitates the qualitative understanding of the contributions of this research.
Answer: Thank you
The authors incorporate 3 self-citations, which in view of the sample analysed and the time frame of the interventions, appear to derive from the same research project. However, in each of the 3 publications they carry out complementary and different analyses. They are relevant because they lend strength to the results reported in the manuscript.
Answer: Thank you
The conclusions are presented in a concrete, clear and precise manner. The authors suggest that vigorous group exercise as a treatment for depression may be valuable for adolescents in the long term.’
Answer: Thank you
A specific section is included with strengths and limitations. For future research it would be interesting to have a larger sample to provide the results with greater scientific soundness.
Answer: We agree
Reviewer 3 Report
An interesting manuscript is presented for publication in the journal, whose objective can be considered original and of interest to the readers of the journal, so it would be good to consider it for publication after the authors make those modifications indicated below. Modifications and improvement suggestions:
List of authors and affiliations: The full names and surnames of the authors must be provided. Initials of any middle name may be added.
Abstract: Authors are urged to revise their wording as there are sentences that are not well understood or are cut off and lack full meaning.
Methodology: It would be interesting if, in complementary material, the authors included the consolidated criteria for the presentation of qualitative research reports (COREQ) or in the text they included the most relevant ones and that they have followed when developing the manuscript.
Discussion: Line 404: Remove the word results Strengths and Limitations: This section should be reformulated in terms of its wording, highlighting the strengths and limitations, avoiding conceptual definitions.
Bibliography: The authors must review this section and put the title of the journals in abbreviated form, according to the indications of the journal itself.
Author Response
We thank the reviewer for a thorough review and helpful comments to improve the manuscript.
Please find the reviewers’ original comments and our response below (the latter using italics). In the revised manuscript, changes have been marked by track change to identify corrections.
An interesting manuscript is presented for publication in the journal, whose objective can be considered original and of interest to the readers of the journal, so it would be good to consider it for publication after the authors make those modifications indicated below. Modifications and improvement suggestions:
Answer: Thank you
List of authors and affiliations: The full names and surnames of the authors must be provided. Initials of any middle name may be added.
Answer: The manuscript has full detail of the authors and affiliations
Abstract: Authors are urged to revise their wording as there are sentences that are not well understood or are cut off and lack full meaning.
Answer: Thank you for the suggestion, we have revised the abstract
Methodology: It would be interesting if, in complementary material, the authors included the consolidated criteria for the presentation of qualitative research reports (COREQ) or in the text they included the most relevant ones and that they have followed when developing the manuscript.
Answer: We have attached a supplementary including COREQ report
Discussion: Line 404: Remove the word results Strengths and Limitations: This section should be reformulated in terms of its wording, highlighting the strengths and limitations, avoiding conceptual definitions.
Answer: We have removed the words: Results and Strenghts and limitations. We have also removed the conceptual definitions. Lines 525, 527-528, 530-531,542-543, 552-553
Bibliography: The authors must review this section and put the title of the journals in abbreviated form, according to the indications of the journal itself.
Answer: Thank you for noticing this. We have reviewed this section and put the titles in abbreviated form.
Reviewer 4 Report
Thank you for the opportunity to review your manuscript
First of all, comment that this manuscript must undergo extensive format and language editing
- Abstract is too long. must comply with the standards of the journal
- In the introduction, the current situation of the subject of study should be explained in greater depth.
- Authors must specify the type of qualitative study
- Authors must specify if the triangulation of the contents of the interviews was carried out.
- The cultural and theoretical orientation of the researcher should be clarified
- The relationship between the researcher and the study participants should be addressed
- The researcher must critically examine their own role and potential influence during data collection
- Authors should be reported how the investigator responded to events that arose during the study
- The discussion section should be further enhanced and enriched
- The conclusion of the study is too long . It should be summarized, taking into account the results obtained
Author Response
We thank the reviewer for a thorough review and helpful comments to improve the manuscript.
Please find the reviewers’ original comments and our response below (the latter using italics). In the revised manuscript, changes have been marked by track change to identify corrections.
Thank you for the opportunity to review your manuscript
First of all, comment that this manuscript must undergo extensive format and language editing
Answer: The manuscript has undergone language editing from a professional native Englishman.
Abstract is too long. must comply with the standards of the journal
Answer. Thank you for noticing this. We have reduced the word count in the abstract
In the introduction, the current situation of the subject of study should be explained in greater depth.
Answer: Thank you, We have explained the current situation with a lack of research in this field. Lines 58-60
Authors must specify the type of qualitative study
Answer: We have used “A qualitative content analysis”. Line 98-103, 169
Authors must specify if the triangulation of the contents of the interviews was carried out.
Answer: No triangulation has been carried out
The cultural and theoretical orientation of the researcher should be clarified
Answer. Thank you for noticing this. Line 170-173 We have added: “To increase trustworthiness, the authors, who had extensive experience in pediatric psychiatry (MD, PhD), nursing (RN, PhD), physiotherapy (PT, PhD) and qualitative methodology, participated in both the research design and data analysis process.”
The relationship between the researcher and the study participants should be addressed
Answer: We have stated. “The interviews were performed by one of the researchers, a nurse with extensive experience in qualitative interviews (IL), who had no previous relationship with the participants.” Line 147-149
The researcher must critically examine their own role and potential influence during data collection
Answer We have clarified this: “The interviewer has extensive experience of conducting qualitative interviews and strived for a constant awareness of having an open mind and listening to the participants by asking open-ended questions by prompting the participants to develop their narrative.” Line 149-151
Authors should be reported how the investigator responded to events that arose during the study
Answer: We have added this “Individual, semi-structured follow-up interviews were conducted during March and April 2019 and took place in a private room at the CAP clinic, where competence was available to take care of any feelings that arose during the interview.” Line 145-147
The discussion section should be further enhanced and enriched
Answer: Since none of the other three reviewers have any concerns about the discussion section, we are unsure about what is expected of us. We have tried to improve the discussion section Lines 534-537, 546-551
The conclusion of the study is too long. It should be summarized, taking into account the results obtained
Answer: We have reduced the text in the conclusion
Round 2
Reviewer 4 Report
Thanks to the authors for taking into account most of my suggestions.
There are still some aspects that should be improved, since they affect the methodological quality of the work:
- The triangulation process is essential in qualitative studies, so the authors should have done it during the study. They must give a more convincing reason why they have not done it. Triangulation would undoubtedly have improved the quality of the study
- Regarding the discussion, the fact that no other reviewer has made any suggestions is not a reason to disregard the one made by me.
Author Response
We thank the reviewer for a thorough review and comments to improve the manuscript.
Please find the reviewers’ original comments and our response below (the latter using italics). In the revised manuscript, changes have been marked by red text to identify corrections.
Thanks to the authors for taking into account most of my suggestions.
There are still some aspects that should be improved, since they affect the methodological quality of the work:
- The triangulation process is essential in qualitative studies, so the authors should have done it during the study. They must give a more convincing reason why they have not done it. Triangulation would undoubtedly have improved the quality of the study
Answer: According to Natow, 2020 triangulation there are five different forms of triangulation: 1. Multiple data sources, 2, Multiple methodologies, 3, Multiple qualitative data analysis techniques, 4, Multiple researchers and 5. Multiple forms of triangulation.
To increase the quality of our study, we have used two different forms of triangulation according to Natow, 2020: 3. Multiple qualitative data analysis techniques with both a deductive and an inductive part and 4. Multiple researchers were involved in different parts and stages in the data analysis to enhance to quality of data in the study. Lines 165-185, 520-521
References: Natow, R. S. (2020). The use of triangulation in qualitative studies employing elite interviews. Qualitative research, 20(2), 160-173.
- Regarding the discussion, the fact that no other reviewer has made any suggestions is not a reason to disregard the one made by me.
Answer: We have now revised the discussion and hope it will be to your satisfaction.